# Morphological Description and Molecular Analysis of Newly Recorded Asteroid, *Henricia djakonovi* Chichvarkhin, 2017 (Asteroidea: Spinulosida: Echinasteridae), from Dokdo Island, Korea

Michael Dadole Ubagan [1], Mariya Shihab Ahmed Alboasud [2] and Taekjun Lee [1,2,*]

1   Marine Biological Resources Institute, Sahmyook University, Seoul 01795, Republic of Korea
2   Department of Animal Resources Science, Sahmyook University, Seoul 01795, Republic of Korea
*   Correspondence: leetj@syu.ac.kr; Tel.: +82-2-3399-1751

**Abstract:** We recently collected the samples of *Henricia* from adjacent waters of Dokdo Island, Korea, using trimix SCUBA diving. Based on a combined result of morphological and molecular analysis, we identified our specimen as *Henricia djakonovi* Chichvarkhin, 2017, which is newly recorded in Korea. Morphologically, *H. djakonovi* has crescent abactinal plates bearing numerous pillar-shaped abactinal spines with a droplet-like apical tip. Moreover, molecular analysis based on the mitochondrial COI gene occurred that clearly distinguished *H. djakonovi* from other species of *Henricia* in the pairwise genetic distance and maximum likelihood analysis. Accordingly, 15 species of *Henricia* are recorded in Korean fauna, including *H. djakonovi*.

**Keywords:** taxonomy; morphology; DNA barcoding; COI

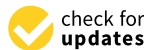



## 1. Introduction

The family Echinasteridae Verrill, 1867 is the only family in the order Spinulosida Perrier, 1884 of the class Asteroidea de Blainville, 1830. This is a small but significant family with just eight genera and a variety of taxa assigned to it [1]. Echinasterids are epifaunal animals that live on the seabed, rock, sand, gravel, or mud at a depth of 6 to 904 m [2]. The genus *Echinaster* Müller and Troschel, 1840 and *Henricia* Gray, 1840 are the genera in which the majority of the species are found. *Henricia* is found mostly in cold waters, including polar habitats and abyssal locations [3]. In the north Pacific Ocean, most of the echinasterids, including *Henricia*, are directly associated with the huge encrusting sponge fauna [4]. *Henricia* in the Atlantic are usually found in rocky subtidal zones [5]. The global diversity of sea stars is believed to be considerably underestimated due to taxonomic issues and sampling biases [6]. In three coastal regions (East Sea, Korea Strait, and Yellow Sea) of Korea, *Henricia* diversity patterns show a higher species richness in the East Sea and lower richness levels recorded in the Yellow Sea [7,8]. A total of 14 *Henricia* species have been reported in South Korea, namely, *Henricia anomala* Hayashi, 1973; *Henricia elachys* Clark & Jewett, 2010; *Henricia epiphysialis* Ubagan, Lee, Kim & Shin, 2020; *Henricia hayashii* Djakonov, 1961; *Henricia leviuscula* Stimpson, 1857; *Henricia nipponica* Uchida, 1928; *Henricia oculata* Pennant, 1777; *Henricia ohshimai* Hayashi, 1935; *Henricia pachyderma* Hayashi, 1940; *Henricia pacifica* Hayashi, 1940; *Henricia perforata* (O.F. Müller, 1776); *Henricia regularis* Hayashi, 1940; *Henricia reniossa* Hayashi, 1940; and *Henricia sanguinolenta* O.F. Müller, 1776 [7,9–12]. According to studies of previous taxonomists, identifying species of Echinasteridae is particularly difficult for a variety of reasons, including: (1) the characteristics of these sea stars are highly variable; (2) they are capable of interbreeding with each other where they overlap to produce intermediates; and (3) they are capable of interbreeding [13,14].

DNA barcoding sequence variation in a 658-base pair (bp) region of the mitochondrial cytochrome *c* oxidase subunit I (COI) gene is a potent tool for the identification and

discovery of species [15,16]. The region of the COI sequence has been validated as an efficient tool for species discrimination in echinoderms [16–18]. This research aimed to add knowledge to the taxonomy of the *Henricia* species in South Korea and the surrounding regions by providing a complete morphological description and molecular analysis of *H. djakonovi*.

## 2. Materials and Methods

A specimen of *Henricia* was collected by trimix SCUBA diving from a depth of 33.7 m on Dokdo Island of Korea on 8 July 2022. The collected specimen was preserved in 95% ethyl alcohol solution immediately. Morphological characteristics such as the size of the disk, the upper and proximal portions of arms, the number of abactinal spines, the shapes of abactinal and actinal skeletons, and the number of adambulacral spines were examined. The morphological features of the specimen were photographed using a scanning electron microscope (JSM-6510; JEOL Ltd., Tokyo, Japan), a stereomicroscope (Nikon SMZ1000; Nikon Co., Tokyo, Japan), and a digital camera (Nikon D7000). The abbreviations used for measurements were the same as those used by Ubagan and Shin [19].

Total genomic DNA was isolated from ethanol-preserved tube feet tissues using a DNeasy blood and tissue DNA isolation kit (Qiagen, Hilden, Germany) according to the manufacturer's instructions. Genomic DNA quality and concentration were determined using a Nanodrop One-C spectrophotometer (Thermo Fisher Scientific, Waltham, MA, USA). All genomic DNA samples were stored at $-20$ °C until further use. The partial sequence of the mitochondrial COI gene (658 bp) was amplified using a pair of primers conserved in echinoderms, Hen_LCO1490m (unpublished) and Hen_HCO2198m (unpublished). PCR was performed using an AccuPower PCR PreMix kit (Bioneer, Daejeon, Republic of Korea) in a total volume of 20 µL in accordance with the manufacturer's instructions. After 1.5 µL of template DNA, 1 µL of each primer at 10 pmol, and 16.5 µL of distilled water were added to AccuPower PCR PreMix, polymerase chain reaction (PCR) was performed with the following amplification parameters: an initial denaturation at 94 °C for 3 min, 35 cycles of denaturation at 94 °C for 30 s, annealing at 48 °C for 45 s, and extension at 72 °C for 1 min, followed by a final extension step at 72 °C for 7 min. The PCR product quality was determined by 1.5% agarose gel electrophoresis staining followed by staining with an EcoDye$^{TM}$ Nucleic Acid Staining Solution (Biofact, Daejeon, Republic of Korea). PCR products were directly sequenced in both directions using an ABI Big Dye Terminator kit and an ABI 3730XL DNA Analyzer (Applied Biosystems, Foster City, CA, USA).

Mitochondrial COI sequences were assembled with Geneious R11 (Biomatters Limited, Auckland, New Zealand). Pairwise genetic distances were calculated using MEGA 7 [20] with the Kimura-2 parameter genetic distances model [21]. Gaps and missing data were completely deleted. The best-fit model of nucleotide substitution for the COI dataset was selected by jModelTest v. 2.1.1 [22] using the Akaike Information Criterion for Maximum Likelihood (ML). The ML tree was constructed using PhyML 3.0 [23] under the GTR+I+G model for the COI dataset. Bootstrap analysis was performed with 1000 replicates, the proportion of invariable sites was 0.487, and the gamma distribution parameter was 0.742. For aligned COI sequences (420 bp), 10 species of *Henricia* were used including *H. djakonovi*. Two species of genus *Luidia* Forbes, 1839 were used as an outgroup.

## 3. Systematic Accounts

Phylum Echinodermata Klein, 1778
Class Asteroidea de Blainville, 1830
Order Spinulosida Perrier, 1884
Family Echinasteridae Verrill, 1870
Genus *Henricia* Gray, 1840
*Henricia djakonovi* Chichvarkhin, 2017 Figures 1 and 2.

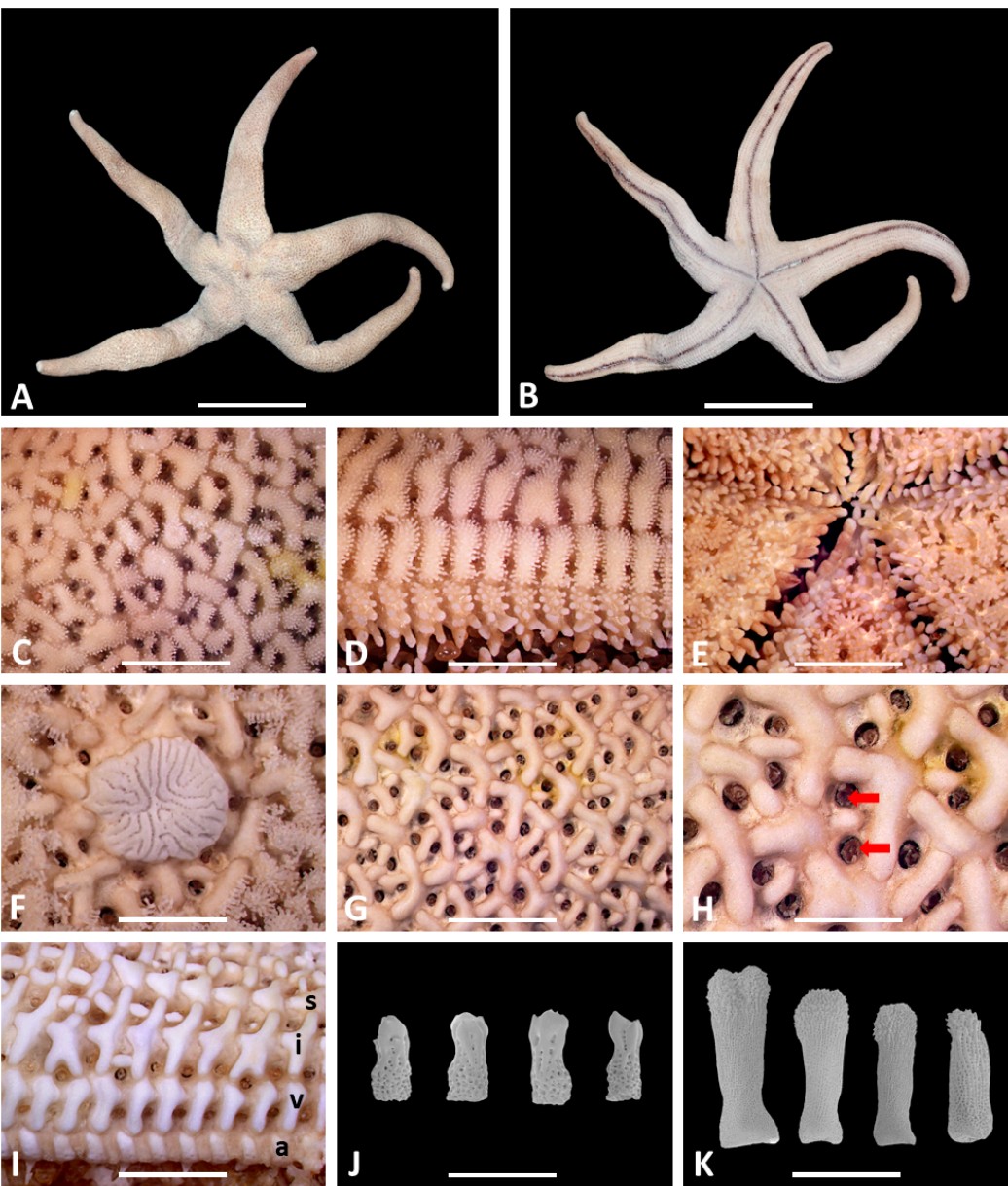

**Figure 1.** *Henricia djakonovi* Chichvarkhin, 2017. (**A**). abactinal side; (**B**). actinal side; (**C**). abactinal paxillae; (**D,K**). adambulacral spines; (**E**). oral part; (**F**). madreporite; (**G**). abactinal skeleton; (**H**). papulae (arrows); (**I**). actinal skeleton: superomarginal plates (s); inferomarginal plates (i), ventrolateral plates (v), adambulacral plates (a); and (**J**). abactinal spines. Scale bars: **A,B** = 2 cm, **C–I** = 1 mm, **J** = 100 μm, **K** = 500 μm (**J**, **K**, SEM images).

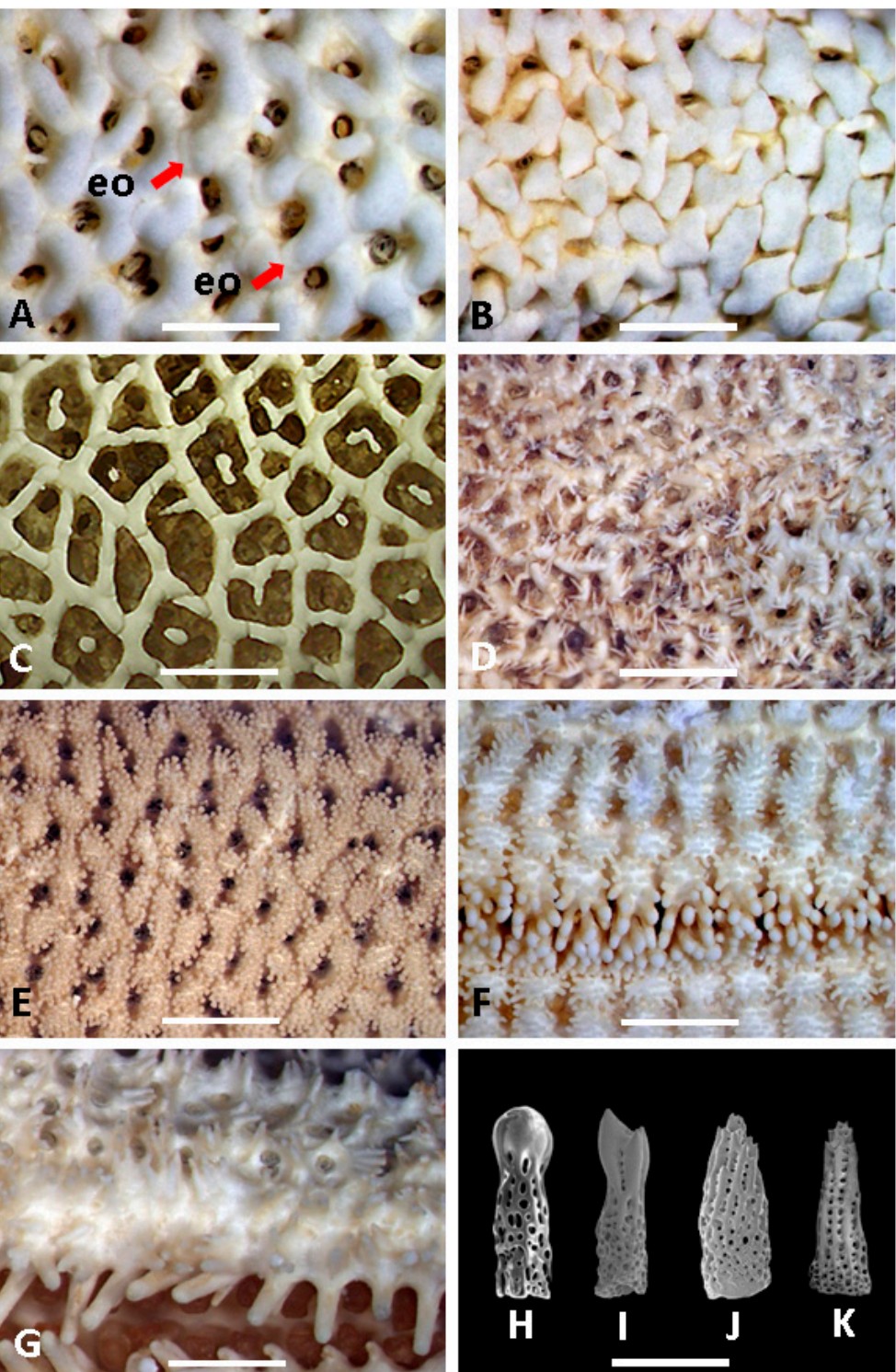

**Figure 2.** Important morphological characters used in the identification of Korean *Henricia* species, denuded abactinal skeleton (imbricated), (**A**). *H. epiphysialis* (eo, extended ossicles), (**B**). *H. regularis*; denuded abactinal skeleton (reticulated), (**C**). *H. pachyderma*; abactinal paxillae: (**D**). *H. anomala*, (**E**). *H. reniossa*; adambulacral spines: (**F**). *H. sanguinolenta* (more than 10 spines), (**G**). *H. ohshimai* (5 or 6 spines); denuded abactinal spines: (**H**). *H. leviuscula*, (**I**). *H. djakonovi*, (**J**). *H. pachyderma*, (**K**). *H. perforata*. Scale bars: **A–G** = 1 mm, **H–K** = 50 μm (SEM images).

*Henricia djakonovi* Chichvarkhin, 2017: 1 [13]; Mah, 2022: 954610 [24].

**Material examined.** One specimen, Dokdo Island: Ulleung-gun: Gyeongsangbuk-do, Korea (37°16′44.3″ N, 131°51′59.3″ E), 8 July 2022, Lee T., a depth of 33.7 m, a temperature of 16 °C, collected by trimix SCUBA diving, deposited in NIBR (FXVZIV0000000202).

**Description.** Arms five, long, large disk, gradually tapering to tips (Figure 1A,B). Abactinal paxillae forming elongated, crescent-shaped with smaller paxillae bearing seven to 18 spinelets; larger paxillae bearing 126–144 robust spinelets (Figures 1C and 2I). Denuded abactinal spines pillar formed with droplet-like apical tip (Figure 1J). Paxillae on the lateral side of arms similar to abactinal paxillae. Abactinal skeleton closely meshed, crescent-shaped, closely united into fine meshed network, abactinal plates larger than papular areas (Figure 1G). Papular areas narrow with one to three papulae in an area (Figure 1H). Some papular areas divided by small ossicles. Madreporite circular in form, slightly sunken, bearing spines similar to adjacent spines (Figure 1F). Three regular series of plates adjacent to adambulacral (superomarginal, inferomarginal, and ventrolateral) plates well defined. Superomarginal forming elongated cross, reaching tip of arm. Inferomarginal plates longer than superomarginal and ventrolateral plates, bearing 130–175 spines, reaching tip of arm. Ventrolateral plates rounded cross-shaped, bearing 55–78 spines, reaching tip of arm. Actinal papular areas bearing one or two papulae. Adambulacral plates forming semi-rounded shape, bearing 13–16 spines, flat tip spines near furrow edge, stout spines near ventrolateral spines, arranged in two transverse series (Figure 1D,I,K). Furrow spine single. Oral part bearing two slender oral spines, three marginal spines, and four or five sub-oral spines (Figure 1E).

**Size.** R = 110 mm, r = 20 mm, R/r = 5.5.

**Habitat.** This specimen inhabited the hard substrates (rocks) of a depth of 33.7 m.

**Distribution.** Korea (Dokdo Island: Ulleung-gun: Gyeongsangbuk-do), Russia (Senkina Shapka pinnacle, Rudnaya Bay).

**Remarks**. *Henricia djakonovi* is easily distinguishable from other *Henricia* species due to its characteristic of possessing spotted live coloration [24]. Morphological analysis showed that Korean *H. djakonovi* differed morphologically from three related *Henricia* species (Table 1): *H. leviuscula*, *H. reniossa*, and *H. sanguinolenta*. Major morphological differences are: (1) shape of abactinal spines (*H. djakonovi*: pillar shape with droplet-like apical tip; *H. leviuscula*: granulliform with solid glassy tip), (2) shape of papular areas and number of papulae (*H. djakonovi*: narrow with 1–3 papulae; *H. sanguinolenta*: wide with 1–5 papulae), and (3) shape of abactinal plates (*H. djakonovi*: crescent and lobed; *H. reniossa*: reniform). Morphological analysis of Korean *H. djakonovi* revealed some morphological variation compared to the holotype specimen. Previously, abactinal plates of *H. djakonovi* have been shown to be crowded with abactinal spines (up to 30 in number) [24]. Our specimen possessed a higher number of abactinal spines (up to 144). However, differences in the number of abactinal spines alone cannot be regarded as a stable character for *Henricia* species identification [25]. Therefore, we consider that Korean *H. djakonovi* is the same species as the holotype specimen. *H. djakonovi* is reported for the first time in the Korean fauna.

**Table 1.** Comparison of the morphological characteristics of *H. djakonovi* with those of related *Henricia* species.

| Characters | *H. djakonovi* (Our Specimen) | *H. leviuscula* (Our Specimen) | *H. reniossa* (Hayashi, 1940) | *H. sanguinolenta* (Our Specimen) |
|---|---|---|---|---|
| Range of R/r (Max R) | 5.5 | 6.0 | 7.8–8.0 | 4.1–4.5 |
| Arm | wide arm base tapering to tip | slender, tapering to tip | slender, tapering to tip | thick arm base, tapering to tip |
| Number of abactinal papula | 1–3 | 1 or 2 | 1 or 2 | 1–5 |
| Shape of abactinal papular area | narrow | narrow | narrow | wide |

**Table 1.** *Cont.*

| Characters | *H. djakonovi* (Our Specimen) | *H. leviuscula* (Our Specimen) | *H. reniossa* (Hayashi, 1940) | *H. sanguinolenta* (Our Specimen) |
|---|---|---|---|---|
| Number of abactinal spine | 126–144 | 40–60 | 40–60 or more | 7–16 |
| Shape of abactinal spine | pillar | granuliform | slender | club shape |
| Shape of abactinal plate | crescent, lobed | roundish or elliptical | reniform | crescent, rod-like |
| Shape of inferomarginal plate | elongated cross, rounded cross | rounded cross, rod-like | elongated cross, rounded cross | elongated cross, rounded cross |
| Number of adambulacral spine | 13–16 | 7–10 | 15–25 | 11–17 |
| Pattern of adambulacral furrow + near ventrolateral plate | 1–3 flat tip + 4–16 shorter | 1 long, slender + 2–10 slender bluntly pointed tip | 1–3 long, slender + 4–25 shorter | 1–3 flat tip + 4–17 shorter, bluntly pointed tip |

Moreover, we obtained a partial sequence of the mitochondrial COI gene and deposited it into GenBank of NCBI (GenBank accession no. OP522340). This is the first registration of the COI data for *H. djakonovi*. Pairwise genetic distances (*p*-distance) were calculated by the Kimura-2 parameter (Table 2). The mean interspecific *p*-distance in *Henricia* was 12.0%, ranging from 5.0% (*H. leviuscula*—*H. sanguinolenta*) to 16.7% (*H. oculata*—*H. regularis*). The mean interspecific *p*-distance between *H. djakonovi* and other *Henricia* species was 13.8%, ranging from 11.0% to 16.4% (Table 2). DNA barcoding analysis of *Henricia* in a previous study revealed that the interspecific *p*-distance was relatively high (mean: 13.7%, range: 2.6–18.3%) but the intraspecific *p*-distance was relatively low (mean: 0.3%, range: 0.0–1.6%) [26]. The *p*-distance showed that the Korean *H. djakonovi* was clearly separated from other *Henricia* species (Table 2). The molecular analysis clearly distinguished *H. djakonovi* from other *Henricia* species. Thus, the mitochondrial COI gene is especially useful and effective for DNA barcoding analyses of *Henricia* species. The maximum likelihood tree also showed that *H. djakonovi* was clearly separated from other species of *Henricia* (Figure 3). The values of bootstrapping were used to indicate support for stable groupings and assess differences between the resulting topologies. A clade of *Henricia* in the ML tree was clearly monophyletic. However, polyphyly occurred between *H. reniossa* and *H. nipponica,* which has a low bootstrap value, 26% (Figure 3). The results of the molecular analyses strongly supported that our *Henricia* specimen is clearly different from other species of *Henricia*. Thus, we newly reported a sea star, *H. djakonovi*, in Korean fauna based on morphology and molecular analyses.

**Table 2.** Pairwise genetic distances (%) within 12 asteroids, comprising 10 species of *Henricia* and two species of *Luidia*, from South Korea and GenBank, based on the Kimura 2-parameter model.

| | Species | GenBank Accession No. | 1 | 2 | 3 | 4 | 5 | 6 | 7 | 8 | 9 | 10 | 11 | 12 | References |
|---|---|---|---|---|---|---|---|---|---|---|---|---|---|---|---|
| 1 | *H. djakonovi* | OP522340 | | | | | | | | | | | | | Present study |
| 2 | *H. epiphysialis* | MT086587 | 13.8 | | | | | | | | | | | | [8] |
| 3 | *H. leviuscula* | LC336732 | 13.3 | 6.4 | | | | | | | | | | | [27] |
| 4 | *H. nipponica* | LC336733 | 15.0 | 9.5 | 10.2 | | | | | | | | | | [27] |
| 5 | *H. oculata* | KT268151 | 13.3 | 14.5 | 14.5 | 15.7 | | | | | | | | | [28] |
| 6 | *H. perforata* | MG934907 | 15.0 | 13.8 | 13.6 | 15.0 | 10.2 | | | | | | | | unpublished |
| 7 | *H. regularis* | LC336739 | 14.0 | 8.6 | 8.1 | 10.2 | 16.7 | 14.8 | | | | | | | [27] |
| 8 | *H. reniossa* | LC336740 | 11.0 | 7.1 | 6.0 | 8.6 | 14.5 | 13.1 | 7.9 | | | | | | [27] |
| 9 | *H. reticulata* | LC336737 | 16.4 | 14.3 | 14.3 | 16.4 | 11.0 | 12.6 | 15.5 | 15.0 | | | | | [27] |
| 10 | *H. sanguinolenta* | KT268150 | 12.6 | 6.0 | 5.0 | 9.8 | 15.5 | 14.3 | 7.1 | 5.2 | 15.0 | | | | [28] |
| 11 | *L. avicularia* | KY305010 | 24.3 | 21.9 | 21.2 | 20.5 | 24.5 | 24.3 | 21.4 | 21.4 | 22.6 | 21.4 | | | [29] |
| 12 | *L. quinaria* | JQ740614 | 24.5 | 22.1 | 21.4 | 20.7 | 24.8 | 24.5 | 21.7 | 21.7 | 22.9 | 21.7 | 0.2 | | [30] |

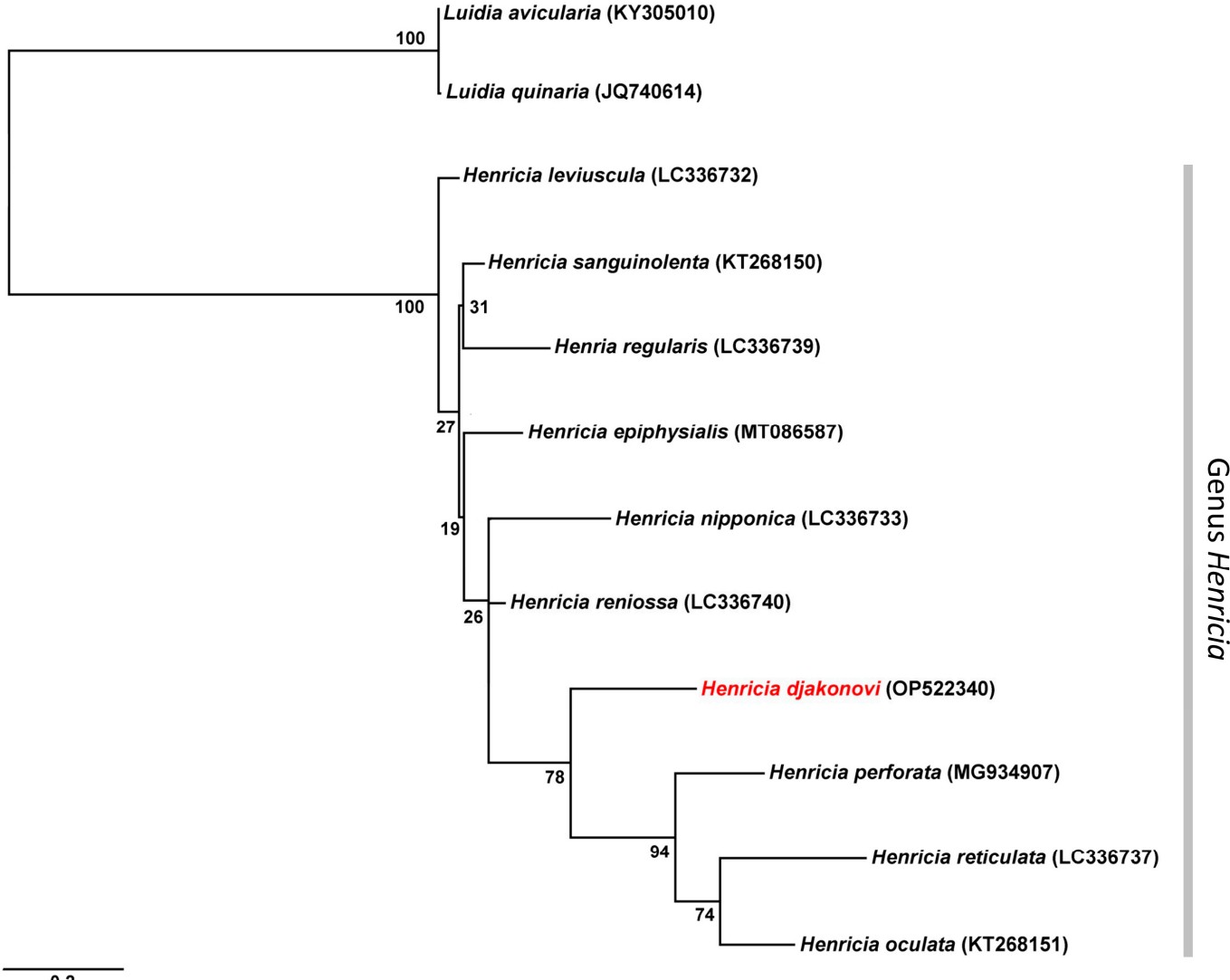

**Figure 3.** The maximum likelihood tree of 10 species of *Henricia* based on the cytochrome *c* oxidase subunit I (COI) dataset (420 bp). *Henricia djakonovi* in this study is marked with red letters. Bootstrap values are indicated on each node.

### Key to species of the genus *Henricia* in Korea

1. Abactinal skeleton generally imbricated in close meshwork ———————————— 2
– Abactinal skeleton generally reticulated in open meshwork ———————————— 9
2. Abactinal and actinal plate extension of ossicles present ——————————— *H. epiphysialis*
– Abactinal and actinal plate extension of ossicles absent ————————————— 3
3. Abactinal paxillae with scattered spinelets, not in distinct group ————————— *H. anomala*
– Abactinal paxillae crowded with spinelets in groups —————————————— 4
4. Shape of abactinal spines pillar ——————————————————————— 5
– Shape of abactinal spines slender ——————————————————————— 6
5. Shape of tip of abactinal spine granular ————————————————— *H. leviuscula*
– Shape of tip of abactinal spine droplet-like ——————————————————*H. djakonovi*
6. Arms long (R/r: >5.5) ————————————————————————— *H. reniossa*
– Arms short (R/r: <4.0) ——————————————————————————— 7
7. Shape of abactinal plates quasi-quadrate —————————————————— *H. regularis*
– Shape of abactinal plates rounded cross ————————————————————— 8
8. Number of adambulacral spines more than 10 ——————————————————— *H. elachys*
– Number of adambulacral spines less than 10 ——————————————————— *H. nipponica*

9. Slender arms; tightly meshed abactinal skeleton ———————————————— 10
– Broad arms; loosely meshed abactinal skeleton ———————————————- 12
10. Adambulacral armature bearing 5 or 6 spines ———————————— *H. ohshimai*
– Adambulacral armature bearing 10 to 18 spines ———————————————— 11
11. Abactinal skeleton strong structured; marginal plates conspicuous ———— *H. hayashii*
– Abactinal skeleton weak structured; marginal plates inconspicuous —————- *H. pacifica*
12. Number of adambulacral spines more than 10 ————————————— *H. sanguinolenta*
– Number of adambulacral spines less than 10 ———————————————————— 13
13. Shape of abactinal spines with broad basal, rapidly taper to tip ———— *H. pachyderma*
– Shape of abactinal spines with narrow basal, slowly taper to tip ——————————— 14
14. Shape of inferomarginal plates reniform ——————————————— *H. oculata*
– Shape of inferomarginal plates rounded cross ——————————————— *H. perforata*

**Author Contributions:** Conceptualization, M.D.U. and T.L.; methodology, M.D.U. and T.L.; validation, T.L.; formal analysis, M.D.U. and T.L.; investigation, M.D.U., M.S.A.A. and T.L.; data curation, M.D.U. and M.S.A.A.; writing—original draft preparation, M.D.U. and M.S.A.A.; writing—review and editing, M.D.U. and T.L.; supervision, T.L.; project administration, T.L.; funding acquisition, T.L. All authors have read and agreed to the published version of the manuscript.

**Funding:** This study was supported by a grant (NIBR202227202) from the National Institute of Biological Resources (NIBR) funded by the Ministry of Environment (MOE), and the Basic Science Research Program through the National Research Foundation of Korea (NRF) funded by the Ministry of Education (2021R1I1A205801) of the Republic of Korea.

**Institutional Review Board Statement:** Not Applicable.

**Informed Consent Statement:** Not Applicable.

**Data Availability Statement:** Not Applicable.

**Conflicts of Interest:** The authors declare no conflict of interest.

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
