# Peer review of "Morphological Description and Molecular Analysis of Newly Recorded Asteroid, Henricia djakonovi Chichvarkhin, 2017 (Asteroidea: Spinulosida: Echinasteridae), from Dokdo Island, Korea"

_2673-6500, doi:10.3390/taxonomy3010004_

Round 1

Reviewer 1 Report

I enjoy reading this manuscript regarding the new record of asteroids from Korea based on morphological and molecular analysis. Actually I am not taxonomist of this group, but can follow the manuscript well. This manuscript is well written but still need to reorganize the structure, add some analyses and recheck some data.

Abstract - I think author need to rewrite and reorganize this part carefully as it is the important part to get reader interested in your full story. Authors probably say in this way:

"We recently collected the samples of the genus Henricia from Dodo Island, South Korea, and based on a combined morphological and molecular analysis, we identified our specimens as H. djkonovi, which is newly record in the country. Morphologically, Henricia djakonovi has … Additionally our molecular analyses based mitochondrial COI gene clearly indicated that… "

Introduction - It would be nice if authors can add some statement about the diversity of this genus in South Korea as this could be important information for reader to follow your story.

Also I don't think authors need to emphasize much more detail in the utilization of COI as bar coding. Mentioning a few sentence is fine. 

M & M - I think authors need to include other COI sequences of H. djakonovi retrieved from NCBI (MG976221, MG976222, KY464038 and  KY464037) into dataset, and then compare with your sequences (OP522340). 

It would be nice if authors reconstruct a phylogenetic tree to look at the relationship of H. djakonovi with other species within the genus.

Results - Frankly I don't think the Table 1 is necessary as authors already mentioned in the results. Intraspecific divergence is much more interesting than interspecific one.  

Dichotomous key to species of Henricia is a bit difficult to follow. Can you amend it as see in Fontoura et al. 2017.

That's all from me.

Good Luck 

Author Response

Dear, Reviewer.

We revised our manuscript and carefully considered all the concerns and issues that were raised by your comments. These changes are described in our point-by-point response below.

Thank you and we look forward to hearing from you again.

Response to the comments of reviewer 1:

This manuscript is well written but still need to reorganize the structure, add some analyses and recheck some data.

  1. Abstract - I think author need to rewrite and reorganize this part carefully as it is the important part to get reader interested in your full story.

Answer:

We rewrite and reorganize the abstract part (Line 9 to 15 of the revised manuscript version).

  1. Introduction - It would be nice if authors can add some statement about the diversity of this genus in South Korea as this could be important information for reader to follow your story.

Answer:

One paragraph is added to the introduction (Line 32-35 of the revised manuscript version)

  1. Also, I don't think authors need to emphasize much more detail in the utilization of COI as barcoding. Mentioning a few sentences is fine.

Answer:

We eliminated some sentences in this concern (Line 50-52 of the older manuscript version).

Response to the comments of reviewer 1 (M&M):
1. I think authors need to include other COI sequences of H. djakonovi retrieved from NCBI (MG976221, MG976222, KY464038 and KY464037) into dataset, and then compare with your sequences (OP522340).

Answer:
Those suggested data come from 16S and 12S ribosomal rRNA in mitochondrial. Thus, we cannot compare those 4 sequences with our study dataset. H. djakonovi has been not registered about DNA barcoding data as mitochondrial COI before. So, that is why we propose firstly COI data of H. djakonovi in this time.

2. It would be nice if authors reconstruct a phylogenetic tree to look at the relationship of H. djakonovi with other species within the genus.

Answer:
We established the maximum likelihood tree and added it to the revised manuscript (Figure 2), follow as your comment.

3. Results - Frankly I don't think the Table 1 is necessary as authors already mentioned in the results. Intraspecific divergence is much more interesting than interspecific one.

Answer:
We agree with your comment. Neverless, we remain the pairwise genetic distance chart in the revised manuscript. Because this study firstly suggests the DNA barcoding data of H. djakonovi, mitochondrial COI. Thus, we want to show that this data clearly initiated and distinctly different from other species of Henricia.
Please, Be understand why we did not follow your wise comment.

Reviewer 2 Report

The investigation is about local problem, but I suppose that species range description and publishing the new sites of finding is important.
In the introduction the authors describe background, but don't write nothing about the aim of the study. Please, improve this part of the article.

Also I recommend to illustrate the distances between the sample and other species by phylogenetic tree, not only with the table of Kimura-2 parameter.

Author Response

Dear, Reviewer.

We revised our manuscript and carefully considered all the concerns and issues that were raised by your comments. These changes are described in our point-by-point response below.

Thank you and we look forward to hearing from you again.

Response to the comments of reviewer 2:

The investigation is about local problem, but I suppose that species range description and publishing the new sites of finding is important.

1) In the introduction the authors describe background, but don't write nothing about the aim of the study. Please, improve this part of the article.

- Answer: We write the aim of the study (Line 39-42 of the revised manuscript version)

2) Also I recommend to illustrate the distances between the sample and other species by phylogenetic tree, not only with the table of Kimura-2 parameter.

- Answer: We executed phylogenetic analysis, and adding a maximum likelihood tree (Figure 2).
